# Residual Model-Based Reinforcement Learning for Physical Dynamics

**Zakariae EL ASRI**
CEDRIC*
zakariae.el-asri@lecnam.net

**Clément RAMBOUR**
CEDRIC*
clement.rambour@lecnam.net

**Nicolas THOME**
ISIR†
nicolas.thome@sorbonne-universite.fr

**Vincent Le Guen**
EDF‡
vincent.le-guen@edf.fr

## Abstract

Dynamic control problems are a prevalent topic in robotics. Deep neural networks have been shown to learn accurately many complex dynamics, but these approaches remain data-inefficient or intractable in some tasks. Rather than learning to reproduce the environment dynamics, traditional control approaches use some physical knowledge to describe the environment's evolution. These approaches do not need many samples to be tuned but suffer from approximations and are not adapted to strong modifications of the environment. In this paper, we introduce a method to learn the parameters of a physical model *i.e.* the parameter of an Ordinary Differential Equation (ODE) to approach at best the observed transitions. This model is completed with a residual data-driven term in charge to reduce the reality gap between simple physical priors and complex environments. We also show that this approach can be naturally extended to the case of the fine-tuning of an implicit physical model trained on simple simulations.

## 1 Introduction & Related Works

Dynamic control problems have been extensively studied in the literature. Prior work handled the subject with physics-based models [13], but these approaches require a high physical knowledge of the problem, a lot of hand engineering, and remain unable to model accurately the dynamics since it's unrealistic to fully describe the true system dynamics *e.g.* friction and contact parameters are either oversimplified or excluded.

Recently, Model-Free deep Reinforcement Learning (MFRL) techniques [16, 19, 8] have shown excellent performances in learning a wide range of control tasks. However, these methods remain data inefficient or even intractable in many real-world scenarios.

Model-based Reinforcement Learning (MBRL) is an interesting alternative since it requires fewer data to learn a dynamic function [7, 5, 10], but is still intractable for some tasks where gathering data is costly or risky. Moreover, an imperfect model can lead to drastic performance degradation when predicting trajectories due to the compounding error effect. Model Predictive Control (MPC) is often used in a closed loop to avoid this kind of drift. MPC consists to plan for a quite long horizon $H$, but we execute only actions on a receding horizon $RH \ll H$. This re-planning has a significant computational cost and represents a challenge in control tasks.

---

*CEDRIC - Conservatoire National des Arts et Métiers, Paris, France.

†ISIR - Sorbonne University, Paris, France

‡EDF R&D, Chatou, France and SINCLAIR AI Lab, Palaiseau, France

Offline Reinforcement Learning Workshop at Neural Information Processing Systems, 2022

To tackle the challenge of data efficiency, further work had been interested in hybrid methods: [20] used MBRL as initialization to accelerate MFRL, [22] presented a method for learning a residual on top of weak policies. Most closely to our work, [1, 15, 11, 17, 12, 24] studied the combination of prior physical knowledge with data-driven residuals.

A recent line of work is focused on developing physic-informed neural networks to model at best complex systems [21]. Among those, models that are based on neural ODEs seem particularly promising to grasp complex physical phenomena and are built to incorporate physical knowledge [4, 2]. APHYNITY [9] aims at augmenting incomplete physical models with fully data-driven corrections.

Our work adapts the APHYNITY framework to MRLB in order to learn the residual and correct the physical parameters governing the approximated physical dynamic. Our approach is mostly physically driven through the integration of a neural ODE describing roughly the environment. It is thus much more sample efficient than fully data-driven model-based methods and manages to be as efficient as data-driven baselines thanks to the learned residual correction. Specifically, our augmented model rises to the two main challenges aforementioned:

- **SAMPLE EFFICIENCY:** It benefits from the sample efficiency of the physical model which is strongly regularized by a given ODE and is as sample efficient as a full data-driven model. Moreover, as our model is able to generalize to complex scenarios, we can use large reseeding horizons without catastrophic drift in the prediction.
- **PERFORMANCE:** It benefits from the data-driven part to compensate for the gap between the approximate model and reality.

This method has proven good performance in the case where the prior is an approximate physical model, we then generalized it to the case where the prior knowledge is a weak model learned on offline simulation dataset.

## 2    Augmenting Physics-Based Model with Data-Driven Residual

We now present our method to learn a data-driven residual on top of a simplified physical model. The overall training scheme is similar to many other MBRL approaches such as [20] and consists in alternating between sequences of model training and model-based control. Following recent work in combining deep learning and physical models [9, 21, 2], the model is composed of a physical model, a prior on the environment's dynamics, and a data-driven residual. The hybrid model can benefit from the regularization expressed by the physical prior and be both sample-efficient and more explainable than a fully data-driven one.

### 2.1    Hybrid Dynamic Model

We assume that the environment can be described by a system following an Ordinary Differential Equation (ODE) of the form:

$$\frac{\mathrm{d}\boldsymbol{s}_t}{\mathrm{d}t}\bigg|_{t=t_0} = F(\boldsymbol{s}_t, \boldsymbol{a}_{t_0}) \tag{1}$$

defined over a finite time interval $[0, T]$, where $\boldsymbol{s}_t$ and $\boldsymbol{a}_t$ are the state and action vector for a given time $t$. The function $F$ is not accessible as it models complex interactions between the agent and the environment. However, we suppose that we can define a physical prior $F_p$ as a rough approximation of the environment's dynamic. The true dynamic is then given by the combination of this prior and a residual: $F = F_r + F_p$.

The two terms composing $F^\theta$ are modeled by two different neural networks $F_p^{\theta_p}$ and $F_r^{\theta_r}$ with parameters $\theta = (\theta_p, \theta_r)$. $F_p^{\theta_p}$ is responsible for correctly identifying the parameters handling the different terms in the equation describing the physical prior while $F_r^{\theta_r}$ purpose is to fill the reality gap on observed transitions. Eq. (1) is solved by integrating the derivative on a trajectory starting in an initial state at time $t$ up to time $t + \Delta t$. A differentiable ODE solver [18] gives the approximate solution:

$$\hat{s}_{t+\Delta t} = s_t + \int_t^{t+\Delta t} (F_p^{\theta_p} + F_r^{\theta_r})(s_{t'}, a_t)\mathrm{d}t' \simeq \mathrm{ODESolve}\big(s_t, a_t, F_p^{\theta_p} + F_r^{\theta_r}, t, t+\Delta t\big) \quad (2)$$

In the end, we want to minimize the error between the prediction $\hat{s}_t$ and the true future state $s_t$.

## 2.2  Training Strategy

Given a set of transitions $\mathcal{D} = \{(s_t, a_t, s_{t+1})\}_{0 \le t \le T}$, we learn the dynamic by minimizing the mean square error (MSE) between the predicted and true future states using stochastic gradient descent. Moreover, as the decomposition $F = F_r + F_p$ is not unique, an $\ell_2$ constraint is imposed over the residual model leading to the following loss:

$$\mathcal{L}(\theta) = \frac{1}{|\mathcal{D}|} \sum_{(s_t, a_t, s_{t+1}) \in \mathcal{D}} \|s_{t+1} - \hat{s}_{t+1}\|_2^2 + \alpha \|F_r^{\theta_r}\|_2$$

$$\text{subject to} \quad \frac{\mathrm{d}\hat{s}_t}{\mathrm{d}t}\Big|_{t=t'} = (F_p^{\theta_p} + F_r^{\theta_r})(s_t, a_{t'}), \quad (3)$$

where $\alpha$ is a coefficient that balances $\|F_r\|_2$ to be as small as possible. The model is learned in a similar fashion as [20]. Random trajectories are first sampled to collect an initial dataset $\mathcal{D}$ which is then used to train $F^\theta$. Afterward, the model-based controller gathers $T$ new transitions and adds them to the training set $\mathcal{D}$, and the model $F^\theta$ is retrained. The algorithm alternates between these two steps of control and data collection on the one hand and training of the model on the second hand until convergence is reached. In this work, we use a simple Cross-Entropy Method (CEM) [6] to select the best sequence of actions $A_t^{(H)} = (a_t, ..., a_{t+H-1})$ over a finite horizon $H$:

$$A_t^{(H)} = \operatorname*{argmax}_{A_t^{(H)}} \sum_{t'=t}^{t+H-1} r(s_{t'}, a_{t'}) \text{ s.t. } \begin{cases} \hat{s}_t = s_t \\ \hat{s}_{t+1} = \mathrm{ODESolve}\big(s_t, a_t, F_p^{\theta_p} + F_r^{\theta_r}, t, t+1\big) \end{cases} \quad (4)$$

# 3  Experimental results

## 3.1  Environments

We evaluated our method on two control benchmark environments from OpenAI Gym [3] classic control tasks: Pendulum ($s_t \in S \subset \mathbb{R}^2$ and $a_t \in \mathbb{R}^1$) and Acrobot ($s_t \in S \subset \mathbb{R}^4$ and $A \in \mathbb{R}^2$). In both tasks, we perform continuous actions to achieve the task of swinging up and balancing the pole on the upright position while the initial state is "hanging down". Parameter values and environment details are listed in Appendix A. In both tasks, we modify the standard environment to have dynamics governed by an ODE that can be represented as the combination of two terms: $F_p$ which the part that can be represented by an analytical model and $F_r$ which represents the random phenomena that cannot be captured analytically. For the differential ODE Solver, we use for all environments the Semi-implicit Euler method [18]. For example, the pendulum dynamic can be described as the combination of a simplistic friction-less model (physical part) and a complex term resulting from friction :

$$s_t = \begin{bmatrix} \varphi \\ \dot{\varphi} \end{bmatrix} \text{ and } F(s_t, a_t) = \begin{bmatrix} \dot{\varphi} \\ \ddot{\varphi} \end{bmatrix} = \begin{bmatrix} \dot{\varphi} \\ C_g \cdot sin(\varphi) + C_i \cdot a_t + C_{Fr} \cdot \dot{\varphi} \end{bmatrix}, \quad (5)$$

where $\varphi$ represents the pendulum's angle, $\dot{\varphi}$ its velocity, $C_g$ is the gravity norm, $C_i$ is the inertia norm and $C_{Fr}$ is the Friction norm.

## 3.2  Experiments

In this work, we used a Multi-Layer Perceptron (MLP) for the data-driven model with ReLu activation (except on the last layer). More parameter details are listed in Appendix B . We run two experiments:

**Experiment 1: Physical model as prior,** where we have a simplistic physical model that approximates the dynamic function (*i.e.* a frictionless model for an environment with friction)

**Experiment 2: Offline learned model as prior,** where we have a model already learned from an environment $Env_A$ and we want to deploy it in an environment $Env_B$. This case can be seen as a Sim-to-Real scenario where $Env_A$ is a simulation and $Env_B$ is the real world.

We consider two baselines for our experiments. First, we show the result of running the initial approximate model without learning. Second, we show the result of learning a model from scratch with a full data-driven MBRL method. We show that combining a prior approximate model and a data-driven residual improves the sample efficiency and the performance of MBRL. We compare also our method with a perfect model that encodes the exact dynamic function.

### 3.3   Results

We present empirical results by comparing our method with the two baselines across the control tasks aforementioned. In the pendulum experiments, the parameters of the MPC are: $H = 80$ the planning Horizon and $RH = 50$ the Receding Horizon for re-planing, while in the acrobot experiment, $H = 30$ and $RH = 10$. We detail in Appendix C an ablation study on the impact of the receding horizon.

For each task, we show that our method combine the best of both models, it benefits from the physical prior to achieve 3-4 x improvement in sample efficiency compared to the full data-driven model in one hand, and benefits from the data-driven residual to gain considerably on performance compared to the physical model.

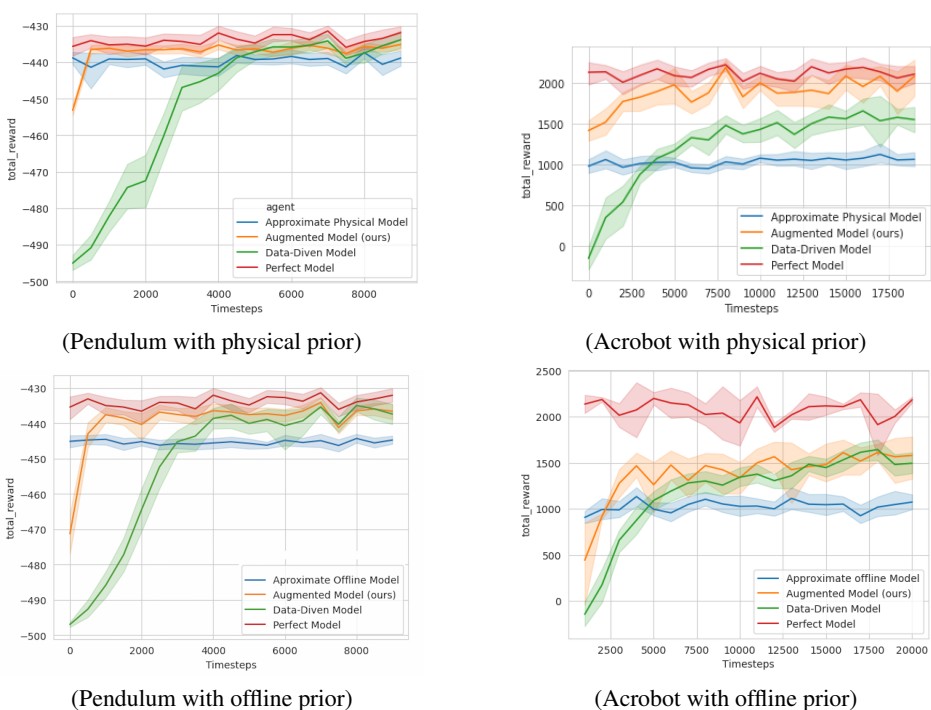

(Pendulum with physical prior)   (Acrobot with physical prior)

(Pendulum with offline prior)   (Acrobot with offline prior)

Figure 1: Plots show the mean and standard deviation of the cumulative reward over multiple runs. Our augmented model shows a 3-4 x improvement in sample efficiency, on both tasks, compared to fully data-driven model, in addition to the improvement in performance compared to the approximate model

## 4 Discussion and Future Work

In this paper, we study an efficient method of augmenting a weak approximate model, either a physical model or a model learned on offline simulations, by a data-driven residual to achieve good performances while remaining data efficient.

We believe that our method can accomplish good result in augmenting prior models in many tasks, especially in the case where the prior model encodes a large part of the dynamic but still need a correction residual. This can be applied in many tasks of robotics that are governed by ODE and need to learn the friction and contact interaction by defining a physical model as prior. Also, this method can be applied as a fast and efficient transfer learning to novel problem scenarios, or adaptation to changes in the environment. Future works would include the integration of our method into more complex dynamics.

## Acknowledgments

We are very grateful to Edouard Leurent for valuable contributions concerning the elaboration of the basis code for this work.

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

# A Environments

In this section, we provide descriptions and details of the environments used in this work. In all environments, the observations are continuous within $[-S_{box}, S_{box}]$ and actions are continuous and restricted to a range $[-a_{max}, a_{max}]$. An overview of both tasks is illustrated in Fig. 2 specific parameters are given in Table 1.

**Pendulum:** A single-linked pendulum is fixed on one end, with an actuator located on the joint. The pendulum is started at the "hanging down" position and the goal is to swing it up and balance it at the upright position. Let $\varphi(t)$ be the joint angle at time $t$. The observation at time $t$ is $(\varphi(t); \dot{\varphi}(t))$.

**Acrobot:** It consists of a pendulum with two links. In this work, we experiment with the fully actuated version of the Acrobot similarly to [25, 23]. Initially, both links point downwards at the "hanging down" position. The goal is to swing up the Acrobot and balance it in the upright position. Let $\varphi_1(t)$ be the joint angles of the first fixed to a hinge at time $t$. and $\varphi_2(t)$ the relative angle between the two links at time $t$. The observation at time $t$ is $(\varphi_1(t); \varphi_2(t); \dot{\varphi}_1(t); \dot{\varphi}_2(t))$.

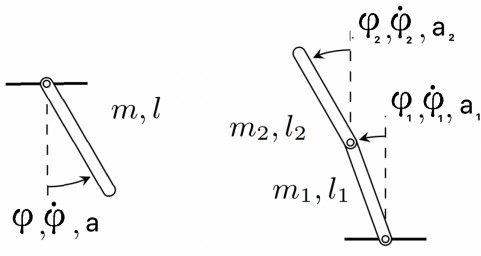

Figure 2: Tasks of our work. Pendulum (left) and Acrobot (right)

Table 1: Environment specifications

| Parameters | Environment | |
| --- | --- | --- |
| | Pendulum | Acrobot |
| Obsevrations space | [ Angle , Velocity ] | [ Angle$_1$, Angle$_2$, Velocity$_1$, Velocity$_2$] |
| $S_{box}$ | $[\pi, \infty]$ | $[\pi, \pi, \infty, \infty]$ |
| Initial state $S_0$ | $[0, 0]$ | $[\pi, 0, 0, 0]$ |
| Actions space | [ Torque ] | [ Torque$_1$, Torque$_2$] |
| $a_{max}$ | $[3]$ | $[3, 3]$ |
| Length of the rollout | 500 Time-steps | 1000 Time-steps |
| $\Delta t$ | 0.02 | 0.02 |

As described in Sec. 3.1, the dynamic functions are the combination of two terms: a physical part and a complex friction part. In our work, we defined for each task two environments: the first is a simulation environment $Env_{sim}$ that encodes only the physical part without frictions, while the second environment $Env_{real}$ encodes the full dynamic function with frictions. In experiment 1, the prior approximate model is a physical model that represents exactly the physical part of the dynamic function. we train our model and the baselines on $Env_{real}$. In experiment 2, we train first a data-driven model on $Env_{sim}$. The learned model will be our prior approximate model, then we train all models on $Env_{real}$

## A.1 Dynamic Functions

In this subsection, we present details of the dynamic function for each task.

For each task, the simulation environment will encode only the physical part (in blue) while the real environment will encode all the dynamic functions (blue + red).

**Pendulum:** Let $s_t = (\varphi; \dot{\varphi})$ be the observation and $a_t$ the action at time $t$. the dynamic of the system is described as:

$$F(\boldsymbol{s}_t, a_t) = \begin{bmatrix} \dot{\varphi} \\ \ddot{\varphi} \end{bmatrix} = \begin{bmatrix} \dot{\varphi} \\ C_g \cdot sin(\varphi) + C_i \cdot a_t + C_{Fr} \cdot \dot{\varphi} \end{bmatrix} \tag{6}$$

Where $C_g$ is the gravity norm, $C_i$ is the inertia norm and $C_{Fr}$ is the Friction norm.

**Acrobot:** Let $s_t = (\varphi_1; \varphi_2; \dot{\varphi}_1; \dot{\varphi}_2)$ be the observation and $a_t = (a_0; a_1)$ the action at time $t$. the dynamic of the system is similar to [25] described as:

$$F(\boldsymbol{s}_t, a_t) = \begin{bmatrix} \dot{\varphi}_1 \\ \dot{\varphi}_2 \\ \ddot{\varphi}_1 \\ \ddot{\varphi}_2 \end{bmatrix} = \begin{bmatrix} \frac{-(\alpha_1 + d_2 + \ddot{\varphi}_2 + \Sigma 1)}{d_1} \\ \frac{\alpha_2 + \frac{d_2}{d_1} \cdot \Sigma_1 - m_2 \times l_1 \cdot lc_2 \times \ddot{\varphi}_1^2 \cdot \sin \varphi_2 - \Sigma_2}{m2 \cdot lc_2{}^2 + I_2 - \frac{d_2{}^2}{d_1}} \end{bmatrix} \tag{7}$$

Where:

$\alpha_1 = a_1 - C_{fr1} \cdot \ddot{\varphi}_1$ such as $C_{fr1}$ is the friction norm in the first joint
$\alpha_2 = a_2 - C_{fr2} \cdot \ddot{\varphi}_2$ such as $C_{fr2}$ is the friction norm in the second joint
$m_1$ and $m_2$ the mass of the first and second links
$l_1$ and $l_2$ the length of the first and second links
$lc_1$ and $lc_2$ the position of the center of mass of the first and second links
$I_1$ and $I_2$ the moment of inertia of the first and second links

$d_1 = m_1 \cdot lc_1{}^2 + m_2 \cdot (l_1{}^2 + lc_2{}^2 + 2 \cdot l_1 \cdot lc_2 \cdot \cos(\varphi_2)) + I_1 + I_2$

$d_2 = m_2 \cdot (lc_2{}^2 + l_1 \cdot lc_2 \cdot \cos(\varphi_2)) + I_2$

$\Sigma_2 = m_2 \cdot lc_2 \cdot g \cdot \cos(\varphi_1 + \varphi_2 - \frac{\pi}{2})$

$\Sigma_1 = m_2 \cdot l_1 \cdot lc_2 \cdot \ddot{\varphi}_2 \cdot \sin(\varphi_2) \cdot (\ddot{\varphi}_2 - 2 \cdot \ddot{\varphi}_1) + (m_1 \cdot lc_1 + m_2 \cdot l_1) \cdot g \cdot \cos(\varphi_1 - \frac{\pi}{2}) + \Sigma_2$

### A.2 Reward Functions

The reward function encodes the desired task. In this work, our main task consists of swinging up and balancing the pole in an upright position. Hence, the height of the pole is an appropriate reward function.

**Pendulum:** Let $s_t = (\varphi; \dot{\varphi})$ be the observation and $a_t$ the action at time $t$. the reward at time $t$ is $\cos(\varphi)$

**Acrobot:** Let $s_t = (\varphi_1; \varphi_2; \dot{\varphi}_1; \dot{\varphi}_2)$ be the observation and $a_t = (a_0; a_1)$ the action at time $t$. the reward at time $t$ is $r_t = 2 - 1.5 \cdot \cos(\varphi_1) - \cos(\varphi_1 + \varphi_2)$

## B  Training Details

In all experiments, we used Adam Optimizer [14] to train our model for 4000 epochs with a learning rate of 0.01. The data-driven model and residual part were parameterized as an MLP with (3 hidden layers for Pendulum and 4 for Acrobot) with 16 neurons at each layer, and ReLu activation in all layers except the last one.

**Offline Model:** The Offline Model is an MLP model (similar to the MLP described above) that was trained in the simulation environment until convergence.

## C  Planing Details

In all experiments, we used CEM [6] as a planner with population size = 300 and elite = 20. For the MPC, we used a planning horizon $H = 80$ and receding horizon $RH = 50$) for the pendulum and ($H = 30; RH = 10$ for the acrobot We studied the impact of changing the receding horizon on the Acrobot Fig. 3.

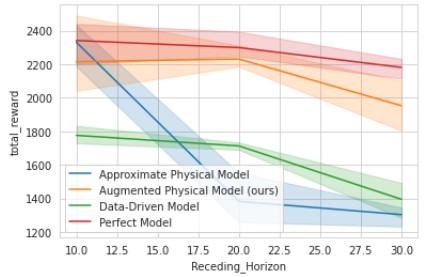
(Acrobot with Physical prior, Horizon = 30)

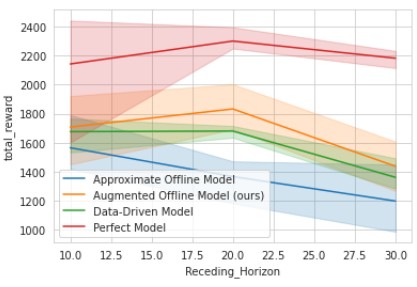
(Acrobot with Offline prior, Horizon = 30)

Figure 3: Plots show the mean and standard deviation of the cumulative reward over multiple runs. Our augmented model shows the best performance when augmenting the receding horizon, compared to fully data-driven model and the approximate physical model

