# OpenReview forum: "Residual Model-Based Reinforcement Learning for Physical Dynamics"
_NeurIPS.cc/2022/Workshop/Offline_RL — Offline RL Workshop NeurIPS 2022_

### Official Review · Reviewer_5FPE · 2022-10-20

**Rating:** 7
**Confidence:** 3

**Review:**

The paper studies the problem of learning dynamics models for control. The proposed approach models the dynamics as an ODE, where the dynamics change is the combination of (1) a physical prior using domain knowledge and (2) a learned residual. The full model is then trained end to end with a differentiable ODE solver.

The method performs better than the fully learned model and just the physics based prior on Mujoco tasks. Overall this seems like a nice way of combining prior knowledge and learning for model-based RL.